# Adsorption Capacity of Vitamin B_12_ and Creatinine on Highly-Mesoporous Activated Carbons Obtained from Lignocellulosic Raw Materials

**DOI:** 10.3390/molecules25133095

**Published:** 2020-07-07

**Authors:** Tudor Lupaşcu, Oleg Petuhov, Nina Ţîmbaliuc, Silvia Cibotaru, Andrei Rotaru

**Affiliations:** 1Institute of Chemistry, Laboratory of Ecological Chemistry, Str. Academiei, Nr. 3, Chişinău, Republic of Moldova; lupascut@gmail.com (T.L.); petuhov.chem@gmail.com (O.P.); timbaliuc_nina@yahoo.com (N.Ţ.); silvia.popovici@gmail.com (S.C.); 2Institute of Geology and Seismology, Str. Gheorghe Asachi, Nr. 60/3, Chişinău, Republic of Moldova; 3Department of Biology and Environmental Engineering, University of Craiova, Str. A.I. Cuza, Nr. 13, 200585 Craiova, Romania; 4Institute of Physical Chemistry “Ilie Murgulescu”, Department of Chemical Thermodynamics, Romanian Academy, Splaiul Independentei, Nr. 202, 060021 Bucharest, Romania

**Keywords:** activated carbons, walnut shells, apple-tree wood, lignocellulosic raw materials, adsorption capacity, highly mesoporous materials, creatinine, vitamin B12

## Abstract

Enterosorbents are widely-used materials for human body detoxification, which function by immobilizing and eliminating endogenous and exogenous toxins. Here, activated carbons, obtained from the lignocellulosic raw vegetal materials of indigenous provenance, have been studied. Walnut shell and wood from local species of nuts and apple-trees were carbonized, and further activated at high temperatures with water vapors in a rotary kiln. A second activation was carried out, in a fluidized bed reactor, but for shorter times. The textural properties of the samples were determined from the adsorption isotherms of nitrogen at 77 K, allowing the obtaining of highly mesoporous materials, while the adsorption capacity permitted an essential rise of six to seven times in the maximal adsorption values of the metabolites, which was determined by the reactivation process. A kinetic study of vitamin B_12_ and creatinine immobilization was performed, the optimal immobilization time for the apple-tree wood reactivated carbons being 2 times longer than for those originating from walnut shells. An additional investigation was also performed in specific conditions that simulate the real environment of immobilization: the temperature of a febrile human body (at the temperature *T* = 38 °C) and the characteristic acidity of the urinary tract and stomach (at the pH of 5.68 and 2.53, respectively). The activated carbonic adsorbents studied here, together with the results of the immobilization studies, show that these procedures can conduct a good incorporation of some endogenous metabolic products, such as vitamin B_12_ and creatinine, therefore presenting a good opportunity for their use as forthcoming commercial enterosorbents.

## 1. Introduction

In the last few decades, the anthropic pollution factors of the environment have increasingly affected the health state of humans [1,2,3]. The increase of exogenous intoxications is one of the most serious consequences of the environmental factors’ degradation. Among the simplest and most efficient methods of human body detoxification is enterosorption [4,5], a procedure for the immobilization and elimination of endogenous and exogenous metabolites from the human body [6,7,8]. Today, this method is largely employed by the majority of health clinics, the removal of poisonous species relying usually on activated carbons, lignocellulosic charcoals, animal bone charcoals, inorganic porous minerals, bone polymeric and silicon-based resins [9,10,11,12,13,14,15,16,17].

The most produced and employed oral intestinal absorbents (enterosorbents) are based on activated carbon; however, their use has to be limited to shorter periods, since they are non-selective adsorbents [10]. The immobilization of various types of contaminants takes place on the activated carbons, mainly due to the physical-chemical properties and adsorption forces within the porous structures. Under these circumstances, the employment of active adsorbents in the complex treatment of intoxications and pathological states within the human organism is favored.

A leading role within the research dedicated to enterosorbents is played by the activated carbons [18] obtained from raw materials of vegetal origin (nutshells, fruit seeds, grape seeds, etc.) [19,20]. The morphological structure of these sources of raw matter favors the obtaining of specimens of activated carbon with high performance values [21] regarding adsorption capacity and increased mechanical resistance [22,23,24]. 

In this work, we study some indigenous, highly mesoporous activated carbons obtained from lignocellulosic raw vegetal materials. The investigation was directed towards the adsorption capacity [10] of these activated carbon materials, which were obtained from local species of nuts and wood (nutshells—*NS* [25,26] and apple-tree wood—*AW* [27]), with respect to the marker samples with medium- and low-molecular masses, such as vitamin B_12_ and creatinine. The carbonaceous absorbents have been obtained in prototype conditions, by following physical-chemical activation procedures. The investigated carbonaceous samples are relatively non-expensive and fairly efficient materials for the immobilization of the marker compounds named above (vitamin B_12_ and creatinine), founding thus the increased interest in their employment as enterosorbents. The porous microstructure of these activated carbons was determined from the adsorption isotherms of nitrogen; also, the influence of a temperature increase and pH lowering on the process of the immobilization of biogenic substances, like vitamin B_12_, was studied. The values obtained for the adsorption of the metabolites correlate well with the mesoporous textural characteristics of the investigated indigenous activated carbons.

## 2. Results and Discussions

The present work fulfills the need to study the obtaining of carbonaceous adsorbents from raw lignocellulosic matter with a developing mesoporous structure; the investigation of biogenic compounds’ adsorption in conditions resembling those of the human body may allow for a better understanding of the influence of pH and temperature on the enterosorption processes. Teixeira et al. [26] have shown that the adsorption of antibiotics into the walnut shell-based activated carbons depends on the temperature and pH of the solution. Therefore, the adsorption capacity increases with the increase of temperature, from 10 to 30 °C, indicating the influential role the diffusion coefficient has upon the adsorption process and the formation of chemical bonds, with the functional groups of the adsorbent. It was also shown that the pH value of the solution may change the adsorption value, as a result of the ionic structure of the adsorbate and of the functional groups present on the surface of the walnut shell-based activated carbons. Shen et al. [28] have also established that increasing the percentage of mesopores in the activated carbon fibers leads to a significant increase in vitamin B12 adsorption.

### 2.1. Textural Study of Activated Carbons

The textural parameters of the investigated indigenous carbonaceous adsorbents presented earlier were determined by the adsorption–desorption N_2_ isotherms, and are shown in Figure 1 (*V*—volume of the adsorbed gas vs. *P/P*_0_—relative pressure). In all isotherms, the hysteresis loops are present, indicating the existence of mesopores in the texture of the activated carbons samples. According to IUPAC (International Union of Pure and Applied Chemistry) classification, all these isotherms are of type IV, exhibiting a mixed pore structure.

The textural parameters calculated from the adsorption isotherms are presented in Table 1. The samples named AC-NS and AC-AW are activated carbons obtained from walnut shells and respectively from apple-tree wood by activation via water vapor of carbonized precursors at 900 °C; the samples named AC-NS-30 and AC-AW-30 are the reactivated versions of the AC-NS and the AC-AW samples for 30 min on a fluidized bed reactor, also in the presence of water vapors at the temperature of 900 °C (the exact experimental conditions are presented in Section 3 of this work). As may be observed from Table 1, after the first activation of the AC-NS and AC-AW samples, the set values for the specific surface area and for the pores’ volumes are reasonable. Indeed, a significant difference cannot be noticed in the textural parameters of the activated carbons obtained either from walnut shells or from apple-tree wood, the specific surface areas of these samples being of about 868 and 801 m^2^·g^−1^, respectively. The reactivity of the AC-NS and AC-AW samples, for 30 min during the activation processes in the fluidized bed (activation with water vapor of the carbonized precursors at 900 °C), was accompanied by a significant increase in the specific surface area, the total volume of the pores and especially the mesopores’ volume. Thus, the specific surface areas increase considerably, up to 1973 and 1760 m^2^·g^−1^, while the total sorptive volumes of the pores, *V_S_*, reach 1.725 and 1.510 cm^3^·g^−1^, for the AC-NS-30 and AC-AW-30 samples, respectively. Since the research aims and the investigations were initiated with the aim of obtaining activated carbons for application in medicine, the study was directed towards realizing a maximal share of the mesopores within the porous structure of these lignocellulosic materials. The data presented in Table 1 indicate that the percentage of the mesopores’ volume, for both materials, is around 70% of the total volume of the pores, which is an expected result since the obtained adsorbents may adsorb molecules of medium and large sizes.

The distribution of the volume of pores (Figure 2) also indicated the change in the morphology of the pores after the second activation: a shift of the maximum for the micropores, and a significant increase in the volume of the mesopores, may be noticed; moreover, pores with dimensions greater than 3 nm also appear, while in the initial samples these were not present. The reactivation leads to the enlargement of micropores that transform into mesopores, and also the occurrence of new pores with larger dimensions; as a result, the microstructure of AC-NS-30- and AC-AW-30-activated carbons becomes predominantly mesoporous.

A detailed analysis of the values of the textural parameters of these adsorbents suggests that the most efficient carbonaceous adsorbent for immobilizing vitamin B_12_ may be the activated carbon AC-NS-30, whose surface characteristics have higher values. However, taking into account the large value of the molecular mass of vitamin B_12_, the greater concentration of mesopores in the AC-AW-30-activated carbon may be the determining feature for the process of its immobilization.

### 2.2. Kinetic Study of Vitamin B12 and Creatinine Retention onto the Indigenous Activated Carbons

One of the most important characteristics of enterosorbents is the time necessary for immobilizing the toxic chemical from the human body. By constructing the curves for the adsorption kinetics of vitamin B_12_, it was possible to determine the retention degree of the marker substance as a function of the contact time in the case of each investigated adsorbent. The experimental results regarding the influence of the contact time [between the adsorbent material and the aqueous solution of vitamin B_12_ (C_0_ = 100 mg·L^−1^)] upon the efficiency of the adsorption process are presented in Figure 3. The retained amount, per unit of absorbent material mass, increases proportionally with the increasing of the contact time between the two phases. During the initial stage (first 24 h), the adsorption process evolves faster, thus 27% (AC-NS) and 20% (AC-AW), and 89% and 71%, of the total amount of metabolite is retained, in the cases of the reactivated adsorbents AC-NS-30 and AC-AW-30, respectively. Furthermore, the adsorption process becomes much slower; the optimal immobilization time of vitamin B_12_ with the activated carbonic adsorbents is 48 h for AC-NS-30 (98%) and 96 h for AC-AW-30 (90%).

In the case of creatinine, the kinetics of the process of adsorption onto the studied lignocellulosic activated carbons (Figure 4), which were determined from aqueous solutions (C_0_ = 100 mg·L^−1^), are much faster; this is because creatinine’s molecular mass is much smaller when compared to vitamin B_12_. Thus, for the establishment of the equilibrium, only 6 h of contact are required. The immobilization capacity of the metabolite onto the AC-NS and AC-AW samples is approximately 50%, while the samples of reactivated adsorbents AC-NS-30 and AC-AW-30 exhibit capacities for retaining creatinine of more than 80%.

### 2.3. Adsorption Study of Vitamin B12 and Creatinine from Aqueous Solutions 

The isotherms for the process of the adsorption of vitamin B_12_ and creatinine onto the activated carbonaceous adsorbents (Figure 5 and Figure 6) highlight the influence of the textural parameters of the adsorbent materials on their capacity for immobilization, and the completion of the adsorption values for each analyzed adsorbed/adsorbent system. The analysis of the obtained data for vitamin B_12_ indicates that during the initial stage, the adsorption rate of vitamin B_12_ is faster onto the AC-NS-30 carbonaceous adsorbent, which has higher values of surface characteristics. Furthermore, the higher percentage of mesopores in the case of the AC-AW-30-activated carbon becomes the determining parameter in the process of vitamin B_12_ immobilization.

For the reactivated carbonaceous samples, the values of the maximal adsorption of vitamin B_12_ range between 238.77 mg·g^−1^, in the case of AC-NS-30, and 295.54 mg·g^−1^ for AC-AW-30, while in the case of the AC-NS and AC-AW samples, these values are about 40 mg·g^−1^. For the adsorption of creatinine, it is worth mentioning that the activated carbon obtained from nutshells exhibits a slightly higher adsorption capacity (68.61 mg·g^−1^) with respect to the sample originating from apple-tree wood (58.16 mg·g^−1^), which is probably due to the higher values of the textural parameters (specific surface area and total volume of pores available for the sorption process). By analyzing the available adsorption data, one may observe that the reactivation process of the carbonaceous adsorbents studied here is accompanied by a substantial increase, of about six- to seven-fold, of the adsorption capacity of vitamin B_12_ and creatinine.

### 2.4. Simulations of Marker Compound Adsorption in Conditions Characteristic of Human Bodies 

In order to have a better model for the possibilities of employing these activated carbons as enterosorbents in conditions characteristic of real human bodies, we have also studied the influence of temperature and pH on the process of the immobilization of biogenic substances like vitamin B_12_. In Figure 7, the adsorption isotherms of vitamin B_12_ from aqueous solutions may be observed at pH 2.35 (in the stomach, the pH has values between 1.35 and 3.5) and at pH 5.68 (for the urinary tract, the pH has values between 5.5 and 6.5). It was found that when raising the temperature up to 38 °C, as well as when diminishing the pH down to 2.35, the immobilization ratio of vitamin B_12_ increases on average 10–15% (Figure 7 and Figure 8).

A fairly important feature of carbonaceous adsorbents, when used for the immobilizing of toxins with medium molecular mass, is the degree of the effective usage of pores, *G*. The volume of pores occupied by molecules of vitamin B_12_, within these activated carbons, may be determined by means of Equation (1):
$$V_{S}=a_{m}\cdot V^{*}$$
where *V_S_* is the volume of pores within the activated carbons that are occupied by molecules of the adsorbate, [*V_S_*] is cm^3^·g^−1^, *a_m_* is the maximal value of the adsorption of the adsorbate into the pores of the activated carbons, [*a_m_*] is mg·g^−1^, *V** is the molar volume of the adsorbent calculated from the values of the Van der Waals lengths and angles between the atoms of the molecules, and [*V**] is cm^3^·g^−1^. 

The value of *V** (the volume of pores within the adsorbents occupied by molecules of vitamin B_12_) was calculated according to the relation in Equation (2) [28], and is 2.94 nm^3^. In Table 2, the results for the estimation of *V_S_* (the volume of pores within the activated carbons occupied by molecules of vitamin B_12_) and *G_mp_* (the degree of effective usage of mesopores) are presented. The obtained data indicate quite high ratios of utilization for the mesopores.
$$G_{mp}=\frac{V_{S}}{V_{mp}}\cdot 100$$

The results point out that the presence of mesopores in the structure of the enterosorbents strongly influences the immobilization kinetics of toxins. By employing activated carbons with high specific surface areas, but with a microporous texture, the effect on the detoxification of the body will take place over a longer period of time, which may diminish the role of the enterosorbent (preventing its physical-chemical properties and side effects from appearing). However, when using mesoporous adsorbents, the diffusion and adsorption of toxins happens faster, and thus the value of the specific surface area becomes a secondary parameter. The adsorbents studied here are capable of incorporating endogenous metabolic products, such as the vitamin B_12_ and creatinine, presenting therefore a high potential for employment as good-quality enterosorbents. Our study correlates very well with recent studies [29,30]; e.g., Li et al. [29] have shown that the quantity of vitamin B12 adsorbed onto lotus root-type activated carbon was 3.7 mg·g^−1^, which was acquired by simulated in vitro hemoperfusion tests. These very good outcomes are representative of the immobilization and removal of middle-molecular toxins form the human blood, indicating the promising application of these biomass adsorbents in medicine.

## 3. Materials and Methods

The activated carbons selected here for the study of their potential medicinal employment are indigenous in provenance, obtained from lignocellulosic vegetable raw materials: walnuts shells (AC-NS, AC-NS-30) and apple-tree wood (AC-AW, AC-AW-30). The activated carbons AC-NS and AC-AW are commercial adsorbents produced by Ecosorbent Company (activated carbon producer from the Republic of Moldova) by activation via water vapor of carbonized precursors at 900 °C (processing time: 2 h, vertical reactor); the AC-AW-30 and AC-NS-30 samples were obtained by reactivation of the AC-NS and the AC-AW samples for 30 min on a fluidized bed reactor, also in the presence of water vapors (temperature = 900 °C, pressure = 1 atm, vertical reactor).

The absorption parameters of these activated carbon materials were determined from the adsorption isotherms of nitrogen at 77 K by means of the AUTOSORB-1MP equipment from Quantachrome [31]. The specific surface area, *S_BET_*, was estimated using the Brunauer–Emmett–Teller (BET) equation. The total volume of pores, *V_S_*, was derived by converting the amount of N_2_ adsorbed gas at the relative pressure of 0.99 into the equivalent liquid volume of the adsorbed material (N_2_). In order to determine the volume of the micropores, *V_μp_*, the “*t method*” (*t*–thin film thickness) was used, while the volume of mesopores, *V_mp_*, was calculated from the difference between the total volume and the volume of micropores. The density functional theory (DFT) method—for the slit/cylindrical pore, NLDFT (Non-Linear Density Function Theory) equilibrium model—was employed for evaluating the effective (predominant) radius of the pores and the distribution of the pores’ volumes as a function of the pores’ radii.

Here, in the role of characteristic metabolites, the vitamin B_12_ (cyanocobalamin, C_63_H_88_CoN_14_O_14_P) with molecular mass of 1355.38 g·mol^−1^ and creatinine (C₄H₇N₃O) with molecular mass of 113.12 g·mol^−1^, were employed. These metabolites are frequently used by researchers as marker compounds for modelling the intoxication state of the organism with medium- and low-molecular mass toxins. 

The absorption processes were studied under static conditions; the adsorption isotherms were measured at room temperature, after the process had equilibrated. The experimental method consists of dosing, with pre-established amounts, the activated carbon, with the constant mass ***m***, with the fixed volume ***V*** of the solutions, which have known concentrations ***C_0_***, while maintaining constant stirring for a reasonable period of time in order to reach the chemical equilibrium. The equilibrium concentrations were determined with the JENWAY 6505 UV-VIS spectrophotometer. The values of the adsorption coefficient for the investigated compounds were obtained according to the relation in Equation (3):$$a=\frac{\left(C_{0}-C_{e}\right)\cdot V}{m}$$
where *a* is adsorption capacity, [*a*] is mg·g^−1^, *C*_0_ is the initial concentration of the solution, [*C*_0_] is mg·L^−1^, *C_e_* is the equilibrium concentration of the solution, [*C_e_*] is mg·L^−1^, *V* is the volume of the solution, [*V*] is L, *m* is the amount of adsorbent and [*m*] is g.

The immobilization investigation that simulates real conditions of the human body has also been performed: the temperature of a febrile human body at T = 38 °C, and the characteristic acidity of the urinary tract, at pH 5.68, and of the stomach, at pH 2.53. The pH of the solutions was measured with the automatic titrometer “Titroline 6000” equipped with an electrode IL-pHT-A120MF, which works in the temperature domain of −5 to 100 °C.

## 4. Conclusions

The adsorption capacities of two marker toxins, which are usual metabolites in the human body (vitamin B_12_ and creatinine), were investigated on highly-mesoporous activated carbons obtained by water-activation and reactivation (via processes undertaken on a fluidized bed) from lignocellulosic raw vegetal materials of indigenous provenance: nutshells and wood from local species of walnuts and apple-trees. The reactivation led to a significant increase of the specific surface area, the total volume of pores, and especially the volume of the mesopores; thus, the textural parameters of the adsorbents represent the decisive factor in the immobilization process of the chosen metabolites.

In general, the walnut shell-activated carbons showed a better retention of metabolites than those of apple-tree wood origin; the essential rise in the maximal adsorption values of the marker toxins vitamin B_12_ and creatinine was determined to be approximately six- to seven-fold, for the case of the reactivated carbons, when compared to the initial activated ones. The presence of mesopores in the structure of the enterosorbents strongly enhances the immobilization kinetics of toxins (the optimal immobilization time in the case of the apple-tree-reactivated carbons is two times longer than the optimal time for those originating from walnut shells).

The parametric study regarding the immobilization of metabolites revealed that the temperature rise, up to 38 °C, and the lowering of the pH down to 2.53, both have a great influence on the process of the immobilization of vitamin B_12_, with the uptake percentage increasing, on average, 10–15%. This proves that the presently investigated enterosorbents are highly efficient when used in experimental circumstances similar to those of interest within the human body (stomach, urinary tract, possible febrile condition).

The activated carbonic adsorbents studied here, together with the results of the parametric studies (reactivation with water vapor, slight temperature increase and considerable pH decrease), show that these procedures are capable of conducting the good incorporation of some endogenous metabolic products, such as vitamin B_12_ and creatinine, which thus presents a good opportunity for these to be used as forthcoming commercial enterosorbents.

## Figures and Tables

**Figure 1 molecules-25-03095-f001:**
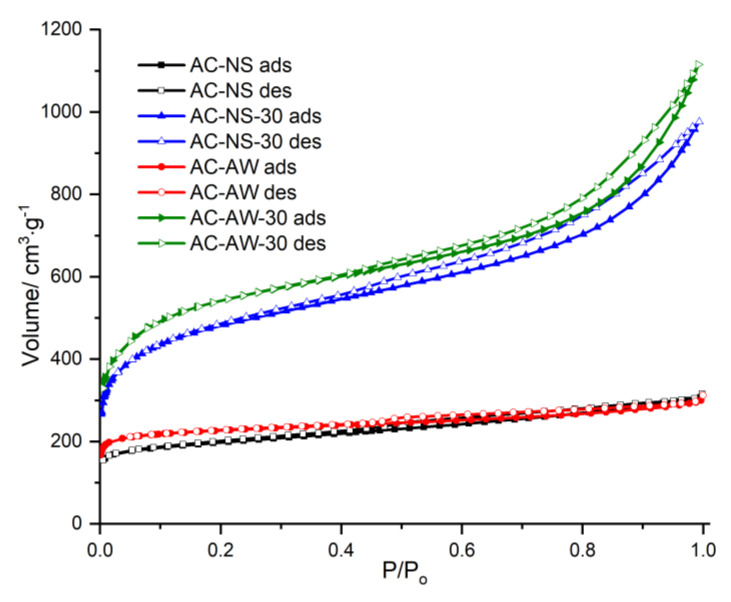
The adsorption–desorption N_2_ isotherms at 77 K for the studied activated carbons.

**Figure 2 molecules-25-03095-f002:**
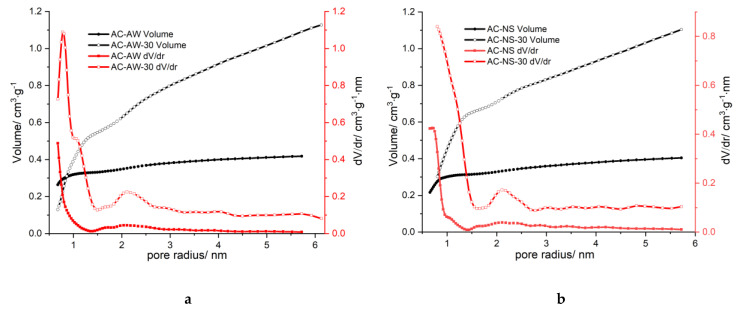
Pore size distributions, *dV/dr*, calculated from (**a**). the N_2_ adsorption–desorption and (**b**). Isotherms on activated carbons, as function of pore radius, *r*.

**Figure 3 molecules-25-03095-f003:**
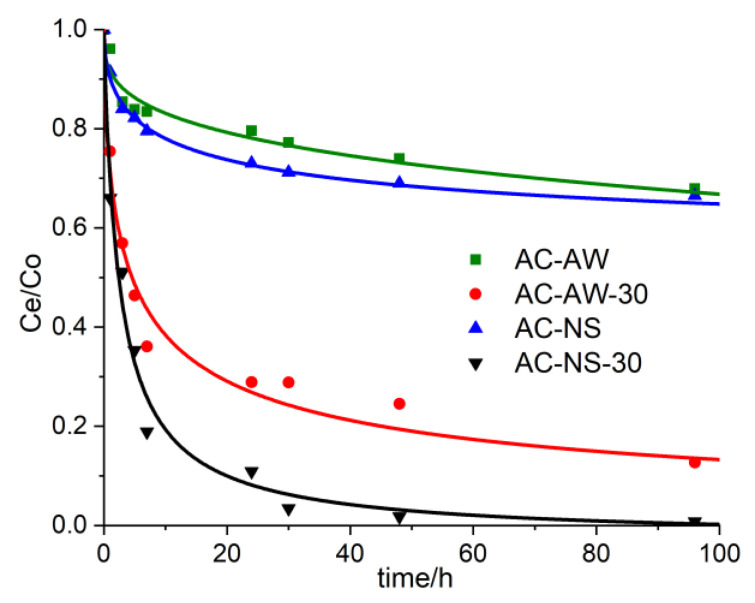
The experimental curves representing the adsorption kinetics of vitamin B_12_ into the activated carbons: AC-AW (green squares), AC-AW-30 (red circles), AC-NS (blue triangles), AC-NS-30 (black triangles).

**Figure 4 molecules-25-03095-f004:**
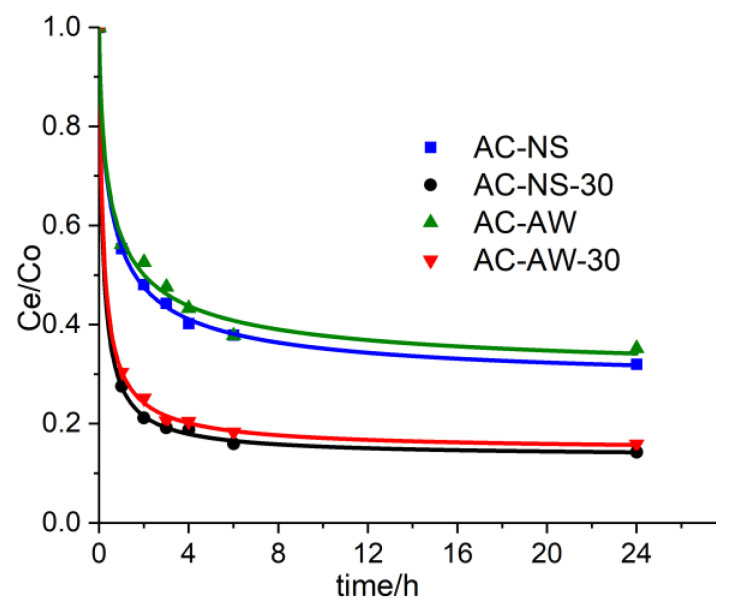
The experimental curves representing the adsorption kinetics of creatinine on the activated carbons: AC-NS (blue squares), AC-NS-30 (black circles), AC-AW (green triangles), AC-AW-30 (red triangles).

**Figure 5 molecules-25-03095-f005:**
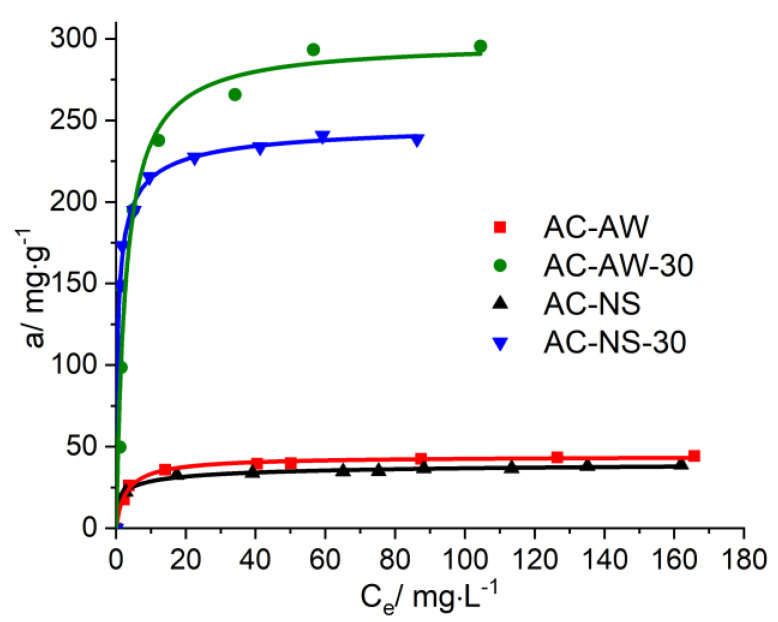
Adsorption isotherms of vitamin B_12_ on AC-NS, AC-NS-30, AC-AW, AC-AW-30.

**Figure 6 molecules-25-03095-f006:**
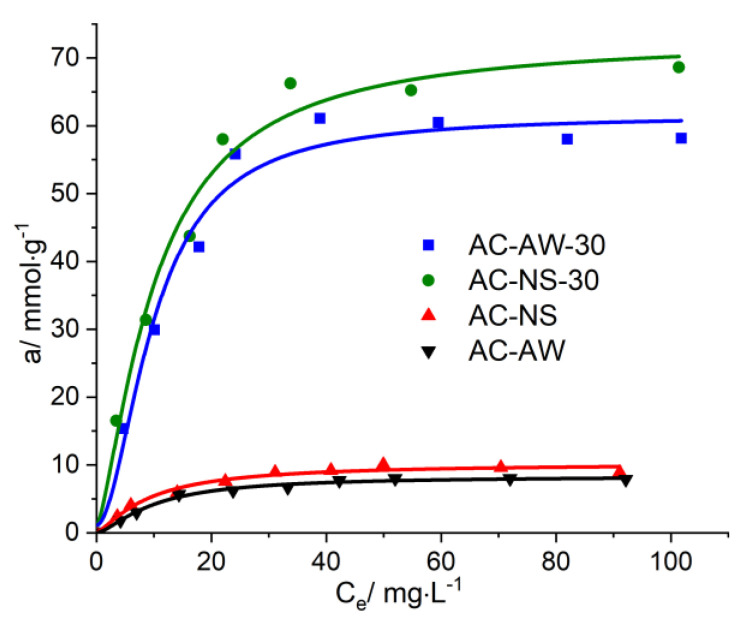
Adsorption isotherms of creatinine on AC-NS, AC-NS-30, AC-AW, AC-AW-30.

**Figure 7 molecules-25-03095-f007:**
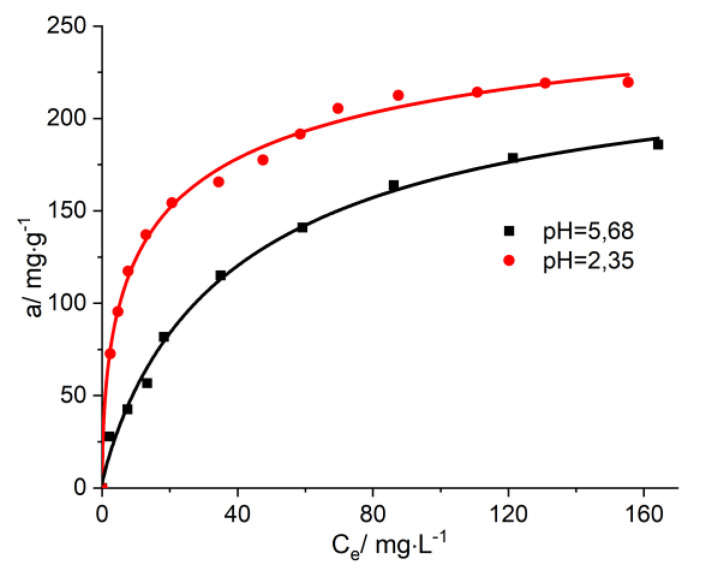
The isotherms of adsorption of vitamin B_12_ onto AC-AW-30 for various pH levels.

**Figure 8 molecules-25-03095-f008:**
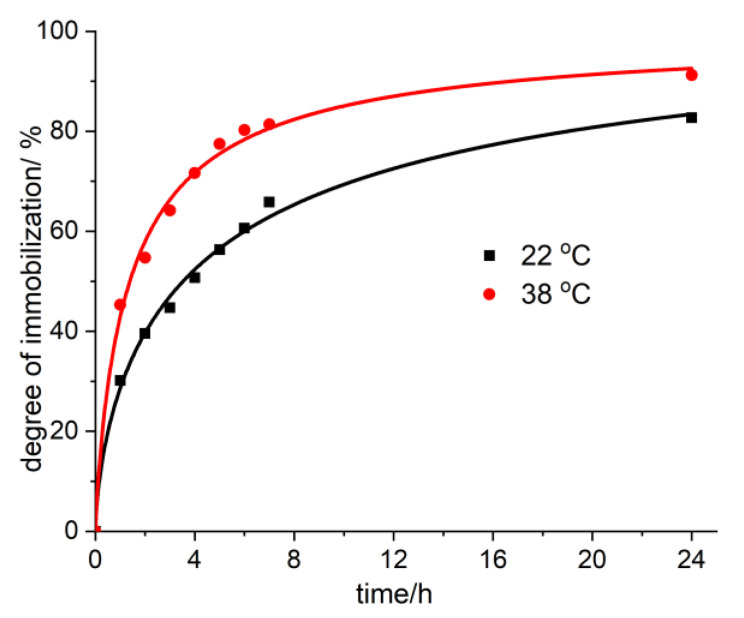
The degree of immobilization of vitamin B_12_ onto AC-AW-30 at 22 °C and 38 °C.

**Table 1 molecules-25-03095-t001:** Textural parameters of activated carbons calculated from nitrogen adsorption–desorption isotherms.

Sample	S_BET_/m^2^·g^−1^	V_S_/cm^3^·g^−1^	V_μp_/cm^3^·g^−1^	V_mp_/cm^3^·g^−1^	Vf_mp_/%
AC-NS	868	0.483	0.287	0.196	40.6
AC-AW	801	0.446	0.263	0.129	28.9
AC-NS-30	1973	1.725	0.548	1.177	68.2
AC-AW-30	1760	1.510	0.443	1.067	70.7

***S_BET_***: specific surface area of pores; ***V_S_***: total volume of pores; ***V_μp_***: volume of micropores; ***V_mp_***: volume of mesopores; ***Vf_mp_***: volume fraction of mesopores.

**Table 2 molecules-25-03095-t002:** The volume of pores occupied by molecules of vitamin B_12_ (*V_S_*), and the degree of effective usage of the mesopores (*G_mp_*) for the activated carbons AC-NS, AC-NS-30, AC-AW, AC-AW-30.

Sample	V_mp_/cm^3^·g^−1^	*a_m_*/mmol·g^−1^	V_S_/cm^3^·g^−1^	G/%
AC-NS	0.196	0.0288	0.0847	43.2
AC-AW	0.129	0.0327	0.0961	74.9
AC-NS-30	1.177	0.1762	0.5180	44.0
AC-AW-30	1.067	0.2182	0.6415	60.1

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
