# Peer review of "Adsorption Capacity of Vitamin B12 and Creatinine on Highly-Mesoporous Activated Carbons Obtained from Lignocellulosic Raw Materials"

_molecules, 2020, doi:10.3390/molecules25133095_

Round 1
Reviewer 1 Report
The manuscript deals with the adsorption capacity of vitamin B12 and creatinine on highly-mesoporous activated carbons from bio-based raw materials. This paper is interesting; however, the merit of the results shall be worked out significantly. The discussion part in the result section have to be highlighted and the results of this study need to discussion with previous results from the literature. The authors should use the references from the list in their manuscript (e.g. Liz et al. 2020, Teixera et al. 2019, Shen et al. 2004, …) for improving this part. Please mention the wood species of the nutshells used (e.g. walnut, black walnut, ….). The title of this manuscript reflects not the whole work and the research task. The manuscript has to be reworked.
Introduction
The read thread is missing. In the last paragraph there is a discussion about important topics, which have to discussed before the last paragraph and also literature should be used for explanations of the mentioned topics. There is not a huge overlapping between the topics from the introduction section and the results sections (e.g. pH-value).
Material and Methods
The authors could divide this section into different subsection, that could improve the readability of this section.
The important process parameters of the second activation are missing (e.g. temperature, pressure, …). Were the parameters the same as in the first activation process, only the process time was different?
How many repetitions on the measurements were done?
Which devices and/or method did the authors use for the pH-value measurements?
Results and Discussion
For the comparison it is easier that all charts in one figure have the same axis scale (Figure 1, and Figure 2). Please check if it is possible;
What do the terms a, b, c, and d mean in Figure 1 and Figure 2? You can mention it in the caption of the figures.
These results shall be discussed with results from the literature.
Conclusions
The conclusion section has to be reworked. Almost all information on the first two paragraphs are not necessary for the conclusions.
Reviewer 2 Report
In this work, the authors have systematically investigated the adsorption performance of B12 and creatinine on four different carbon materials. The adsorption capability is well correlated with pore size and pore volume. Thus, I would recommend acceptance of this work in Molecules after addressing the following issues.
- The significance and novelty of this work is not well justified in the introduction part.
- Line 62: Is B12 a toxic compound?
- In Figure 2: Is the pore size distribution derived from adsorption branch rather than isotherms? What is the method for the pore size distribution? Why the pore size distribution is as function of half pore width not pore diameter? What is dv/dr curve and how to interpret this set of data? Typo in Figure 2 legend: “widith”
- The conclusions part should be concise.
Reviewer 3 Report
The main aspects to be considered, in my opinion, are the following:
The phrasis reported at lines 61-62: “The investigated carbonaceous samples are relatively non-expensive and fairly efficient materials for the immobilization of toxic compounds named above (vitamin B12 and creatinine), presenting thus an increased interest for the employment as enterosorbents” should be stressed by the authors to better specify, the novelty of their work also referring to eventual new publication, if appropriate, of this year (2020 is missing in the references list):
- Introduction: Thus the raw materials are local, less expensive and also of eco-compatible production? In fact, the raw materials studied in this work are reported to be ‘fairly efficient’ in comparison with highly performant enterosorbents already in use, either in term of textural characteristic and ‘non- selectivity’ towards vit. B12 and creatinine, as marker adsorbates of middle and low molecular mass.
- Results: the adsorption and absorption processes of all samples, AC-AW, AC-AW-30 and AC-NS, AC-NS-30, could be better summarized in relation to the different immobilization time and extent of Vit B12 and creatinine, having different sizes and hydrophilicity. Particularly, the experimental details reported in figs 7 and 8 for vit. B12 should be better specified, starting from the legenda. Why rising the temperature up to 38 °C and lowering the pH down to 2.53? Was the initial pH of the vitamin in water measured and why not to use a buffered (entero- suitable) pH for the experiments?
- Finally: it should be mentioned in Conclusions, the utility of acquiring SEM images of surfaces and cross- sections of the samples in order to visualize, at different magnifcation, the surface morphology and the inside distribution of pores.
Few typing errors
Round 2
Reviewer 1 Report
The authors reworked the manuscript according to the reviewer's comments. The results are clearly presented and discussed with some references.